# Nursing Care Plan for Patients with Tay–Sachs—A Rare Paediatric Disease

**DOI:** 10.3390/jpm13081222

**Published:** 2023-08-01

**Authors:** Jonathan Cortés-Martín, Beatriz Piqueras-Sola, Juan Carlos Sánchez-García, Andrés Reinoso-Cobo, Laura Ramos-Petersen, Lourdes Díaz-Rodríguez, Raquel Rodríguez-Blanque

**Affiliations:** 1Research Group CTS1068, Andalusia Research Plan, Junta de Andalucía, Nursing Department, Faculty of Health Sciences, University of Granada, 18071 Granada, Spain; jcortesmartin@ugr.es (J.C.-M.); cldiaz@ugr.es (L.D.-R.); rarobladoc@ugr.es (R.R.-B.); 2Hospital University Virgen de las Nieves, 18014 Granada, Spain; bpiquerassola@gmail.com; 3Department of Nursing and Podiatry, Faculty of Health Sciences, University of Malaga, Arquitecto Francisco Peñalosa 3, Ampliación de Campus de Teatinos, 29071 Malaga, Spain; andreicob@uma.es (A.R.-C.); lauraramos.94@uma.es (L.R.-P.); 4Hospital-Universitariy Clínico San Cecilio, 18016 Granada, Spain

**Keywords:** rare disease, Tay–Sachs, nursing care plan, clinical practice, healthcare, Delphi method

## Abstract

Tay–Sachs disease is classified as a rare paediatric disease of metabolic origin. It is an autosomal recessive inherited disease. The gene responsible for the disease is known as HEXA, and it is located on chromosome 15(15q23). There is currently no effective treatment for Tay–Sachs disease; hence, it is an incurable disease in which patients do not live for more than five years, meaning that nursing care takes on greater importance to maintain quality of life. The main objective of this work is to develop a specific standard nursing care plan by applying an inductive research method supported by nursing methodology using the NANDA-NIC-NOC taxonomy and validated by the Delphi method. This care plan will improve the knowledge of health professionals on this topic and support future studies on the disease. Following its implementation, the care plan proposed in this study aims to increase the quality of life of patients diagnosed with this disease.

## 1. Introduction

Under the concept of rare diseases lies a wide range of very diverse diseases for which there is often a scarcity of knowledge [1].

These diseases can have their onset at any age and can present totally different symptoms, not only from one disease to another but also between patients suffering from the same disease but with different degrees of involvement and evolution [2].

For a disease to be classified as rare, it must have a prevalence of less than 1 in 2000 live births. There are currently more than 7000 rare diseases worldwide, of which only 800 have a minimum of scientific knowledge [3].

The comprehensive care required for patients diagnosed with one of these diseases requires the precise study of the different clinical aspects surrounding this type of disease; the specific circumstances that each disease creates; and the repercussions on the family, the environment, and the patients themselves, as well as the existing relations between the above-mentioned sectors [4].

The main problems encountered in this field of study are chronicity, the degenerative nature of such rare diseases, and the high disabling potential and high mortality rates associated with rare diseases [1]. Therefore, this type of patient will require holistic and multidisciplinary interventions and care.

Another aspect of the scenario that is worth highlighting is the delay in the diagnostic phase caused by the lack of knowledge of these diseases and the clinical overlaps between different diseases [5].

The uncertainty of such patients about the state of their health and their future favours the development of mental disorders such as depression and anxiety [6].

Despite the unquestionable progress made by the scientific community with respect to these diseases, to date, 42.68% of people with these diseases do not receive treatment, or if they do, it is inadequate [3]. The approaches intended to alleviate these diseases are based on treating the corresponding different clinical complications and assessing the needs of each individual in order to provide care that increases quality of life.

Tay–Sachs disease is considered to be a rare paediatric disease that is metabolic in nature, belonging to the group of rare lysosomal diseases [7]. It can also be classified as a lipid storage disease (lipidosis) or a disease of the nervous system. As it is considered a rare disease, it is listed in the ORPHANET database under reference number ORPHA845 [8] and in the OMIM database under reference #272800 [9].

It is an autosomal recessive inherited disease. The gene responsible is known as HEXA, and it is located on chromosome 15(15q23) [10]. It has a prevalence of 1 in 320,000 live births [8]. This disease is more frequent among people with Jewish ancestry, specifically Ashkenazi jews, with a higher incidence among Canadians, French people, and members of the Louisiana Cajun population. In the United States, approximately 1 in 27 Ashkenazi Jews are carriers of this disease. In contrast, the carrier rate in the general population and in Jews of Sephardic (Spanish or Portuguese) origin is approximately 1 in 250 [11].

As described above, Tay–Sachs disease is a lysosomal disease caused by Hexosaminidase-A deficiency, an enzyme involved in the biogenesis of lysosomes [12]. Due to the deficiency of this enzyme, gangliosides accumulate and degenerate in the central nervous system, leading to gangliosidosis. These macromolecules cannot be degraded and accumulate, severely damaging neurons and therefore causing a series of pathophysiological processes that lead to the development of the disease. There are several gangliosides, including GM1, GM2, and GM3, which are differentiated by the type of sugar. In particular, Tay–Sachs disease is caused by GM2 gangliosidosis [10].

The development of the disease is characterised by a wide variety of signs and symptoms, most of which result from the progressive degeneration of the central nervous system [13]. There are three variants of the disease, with each being differentiated by the age of onset of the first symptoms [14].

The infantile form (type 1) begins between 3 and 6 months of age. Newborns, up to six months of age, develop normally but gradually begin to lose their physical and mental capacities and die by the age of 5 years.

In the juvenile form (type 2), the onset of the disease is between 2 and 6 years of age, and death occurs around 15 years of age. The decrease in Hexosaminidase-A activity is less marked than in the infantile form.

The chronic adult form (type 3) may begin around the age of 10 years, although the disorder is often not diagnosed until adulthood.

Of the three types of disease, type 1 is the most common, with the other types being generally less severe.

Tay–Sachs disease presents with a series of signs and symptoms that form a broad clinical spectrum depending on the level of Hexosaminidase A deficiency, the variant of the disease, and the degree of involvement and evolution of each patient [11].

The characteristic signs and symptoms of the disease are as follows:

Delayed mental development due to neurological impairment, leading to effects such as speech impairment; impaired cognitive functioning; and impaired personal, temporal, and spatial orientation and memory. Visual, olfactory, tactile, bodily, auditory, and gustatory functions may also be impaired. Language and attention and executive functions such as working memory, planning and reasoning, and mental and social skills may also be impaired [15]. Alterations at the locomotor level, notably hypotonia, muscle weakness, spasticity, loss of balance, ataxia, hand tremors, abnormal movements, paralysis, and exaggerated motor response to auditory stimuli [16]. Other clinical features of concern include macrocephaly, seizure presentation, deafness, and blindness [17,18].

Upon fundus examination, a cherry-red spot can be seen. This sign may be indicative of the disease, although it is not specific for Tay–Sachs, as it can be found in other lysosomal diseases. The involvement of the macula by this spot causes the progressive loss of one’s vision, leading to blindness [19]. It should be noted that all these symptoms are a consequence of the progressive degenerative state of the nervous system [20].

The initial presentation of the disease can be determined by using a patient’s clinical data. The presentation of some of the signs or symptoms described above would provide an initial indication of the disease [12]. To reach a final diagnosis, several tests, such as the following, are performed [7,13,21]:

Funduscopic examinations are performed to study the state of the macula, the degree of visual impairment, and the presence of the cherry-red spot. Neuroimaging studies are carried out to analyse the level of white matter involvement. Studies of the enzymatic activity of Hexosaminidase A via a blood draw are carried out to analyse leukocytes, or skin biopsies are used to study fibroblasts. Genetic studies of the HEXA gene in prenatal diagnosis are also useful, and they are typically performed at 10 weeks of gestation, collecting a sample of chorionic villus. Amniocentesis, usually performed between 15 and 18 weeks of pregnancy, is another useful test; for this, a needle is inserted into the mother’s abdomen to take a sample of the fluid surrounding the foetus. The fluid contains foetal cells that are tested for the presence of Hexosaminidase A. Another method is preimplantation genetic diagnosis, which is performed by selecting healthy embryos prior to in vitro fertilisation.

In very closed population circles where the rate of consanguinity is very high, such as Ashkenazi Jewish populations, one method of diagnostic prevention is mate choice. The aim of this method is to prevent two carriers of the disease from having offspring with each other [22]. 

Currently, this disease has no specific treatment, making it an incurable disease where patients do not live longer than five years [8]. There is no curative or effective treatment for this disease, and nursing care is of vital importance for the maintenance of the quality of life and well-being of the patient, their family, and their environment. There are no scientific studies on nursing care for patients with Tay–Sachs. The aim of this study is to present a standard and specific care plan for this disease. This care plan would improve knowledge in this field, improve the quality of care by tailoring treatment approaches to the disease, and improve the quality of life of patients and that of their families. These reasons were the justification for carrying out this study.

The creation of a care plan for these patients is crucial since no care plans currently exist. Care plans for this disease should be specific and based on the contemporary knowledge of the disease, not in a generic way that involves tending to the underlying symptoms, as is the case currently.

## 2. Materials and Methods

The research method used to develop this care plan was a logical research method based on the use of reasoning in its functions of deduction, analysis, and synthesis. Specifically, an inductive method was implemented in which, starting from particular cases, general knowledge is developed. This method allows for the formulation of hypotheses, an investigation into scientific laws, and demonstrations. In this case, based on the study of cases of patients diagnosed with Tay–Sachs disease and a review of the scientific literature published on the subject, it was found relevant to draw up a specific standard nursing care plan for this type of patient because no care plans currently exist and it is necessary to improve the clinical approach to this disease. In addition, the inductive method, which is a problem-solving method used by nursing professionals, has been endorsed by nurses, and the present study was evaluated and agreed upon by a group of experts using the Delphi method.

Nursing methodology consisting of the application of the scientific method to the nursing care process was used throughout the preparation of this report [23]. It is a systematic process known as the Nursing Care Process, which is based on providing efficient, logical, and rational care that focuses on achieving expected goals through a series of interventions [24]. It is a structured method that is, in itself, classified as a deductive theory and considered a measure of quality in professional practice, providing scientific status to the profession [25].

The standard care plan for Tay–Sachs disease follows the indications of the Nursing Care Process, which aims to ensure the quality of care and provides a basis for operational control and a means for systematisation and research in this field.

This process consists of five phases: assessment, diagnosis, planning, implementation, and evaluation [23]. Through applying this method, the needs and problems of each patient can be identified, and an attempt to meet their needs and alleviate their problems can be made by offering evidence-based nursing care. Once the problems have been identified, objectives can be established and achieved by using a series of proposed and planned interventions. At all times, the implementation of the care plan should be reviewed and evaluated and adapted to the individual patient.

The development of the last two phases, implementation and evaluation, directly depend on the implementation of the plan and will be different for each individual patient.

Nursing methodology, also known as nursing process or nursing scientific method, is a systematic approach that nurses use to provide individualised, evidence-based healthcare to their patients. It was developed to ensure that nursing care is consistent, well-organised, and patient-centred [25].

It is important to note that the nursing process is cyclical, meaning that the steps are repeated and continually adjusted based on patient progress and outcomes. Nursing methodology is essential to provide quality, evidence-based care, and it enables nurses to make informed, patient-centred decisions throughout the care process [24]. 

The key steps of nursing methodology that were applied to the creation of this care plan are described below.

### 2.1. Assessment

Assessment is the first step in determining a person’s health status. It is the most important part in the development of care plans, as a good initial assessment helps to detect health problems that will later be translated into nursing diagnoses. It is a process based on a strategy for collecting and organising all patient information. In this phase, it is essential gather information on the state of one’s health and to search for evidence of abnormal functioning or risk factors that could contribute to the development of health problems [25].

It is an ongoing process of information exchange between the nurse and the patient. The nurse must obtain the necessary data to detect the patient’s needs through a clinical interview, where the professional will take into account both verbal language (words used, form of expression, tone of voice, etc.) and non-verbal language (looks, gestures, grimaces, etc.). Physical examinations are carried out with the aim of finding more clinical data that may be of help in drawing up their care plan. In addition, the patient’s family and environment should also be considered. The importance of consulting the patient’s clinical history to completely gather all relevant information should be emphasised. Once all the data have been collected, a nursing model is chosen to organise the information [24]. All these theoretical aspects were applied to develop the care plan presented in this study.

### 2.2. Diagnosis

In this second phase of the nursing care process, the aim is to list the problems that arise from the different alterations found in the previous phase. Based on the assessment, the patient’s problems are obtained, which will be transformed into diagnoses using the NANDA taxonomy [26]. The NANDA taxonomy is nothing more than a way of naming each problem in order to convert it into a nursing diagnosis. For nursing professionals, the use of the NANDA taxonomy is essential in the routine practice of their profession. Some advantages of using the NANDA taxonomy include the fact that it allows for the establishment of a common language and universal character for the care plan, the implementation of the PAE (Nursing Care Process) as a working method, and dynamic participation within different health teams.

According to NANDA, nursing diagnoses are clinical judgements regarding the individual, family, or community that derive from a systematic deliberate process of data collection and analysis. They provide the basis for prescriptions for definitive therapy, for which the nurse is responsible. In order to formulate a nursing diagnosis correctly, the diagnostic label (reference number by which the diagnosis is known in the healthcare realm) is not sufficient, as several components must be considered. These components are governed by the PES format (Problem, Aetiology, and Symptoms) [23]. There are 4 types of diagnoses according to NANDA.

Problem-focused diagnosis: Involves describing one’s health state/lifestyle choices and how they may have been affected by their family or community. This type of diagnosis is always validated by the presence of signs and symptoms. Formulation structure: Problem (diagnosis) + related with (r/w) Aetiology + manifested by (e/b) Symptoms.

The potential or risk diagnosis describes human responses to processes that may be relevant to the patient, family, or community. Formulation structure: Problem (diagnosis) + related with (r/w) Aetiology.

Health promotion diagnosis: Clinical assessment of the motivations and desires of a person, family, or community in order to improve their well-being and fulfil their health potential as manifested in their willingness to improve health behaviours. Formulation structure: Problem (diagnosis) + manifested by (e/b) Symptoms.

Syndrome: A group of signs and symptoms that almost always occur together. Together, these groups represent a specific clinical picture. Formulation structure: Problem (diagnosis).

Once the specific diagnoses have been listed, objectives will be proposed that must be met through a series of interventions in order to achieve the desired outcomes. It is this care strategy approach that is developed in the next part of the process, planning. All of these theoretical aspects were applied to develop the care plan presented in this study.

### 2.3. Planning

In a similar way to how the NANDA taxonomy was used in the previous phase to name diagnoses, we will now use the NOC and NIC taxonomies to name goals and inter-venations, respectively [27]. This type of taxonomy is used to give a universal character to the care plan.

In the planning phase, there are four basic activities: determining priorities, setting expected outcomes (NOC) [28], choosing the interventions (IAS) [29] to be prescribed to achieve the objectives, and, finally, developing the care plan.

Once the care plan has been developed, it is expected that the proposed interventions will achieve the set objectives and thus solve the patient’s problems.

Within the interventions, in addition to nursing care, the patient could be treated by other health professionals to treat some alterations in a more specific way. For example, for treating alterations of the locomotor system and their consequences and alterations at a cognitive level, physiotherapy sessions and psychological treatment, respectively, would complement nursing interventions, offering the patient a specific and multidisciplinary treatment option that would contribute to improving their quality of life. All these theoretical aspects were applied to develop the care plan presented in this study. 

### 2.4. Implementation

Once the planning stage has been completed, the plan is put into practice, and the patient’s responses are observed. At this stage it is very important to assess the patient’s condition before and after each intervention and to record the patient’s progress. The implementation process consists of preparing the environment and equipment, frequently recording the patient’s progress, and reviewing the care plan [24].

For this last part, it would be useful to ask questions such as the following: Are the problems still the same? Have any new problems emerged? Should anything be done differently than planned? Are the interventions still appropriate?

In this way, the nursing staff can update the care plan according to the progress of the patient.

It should be noted that this study presents a standard and specific care plan for patients diagnosed with Tay–Sachs. The application of this care plan is expected in the future after the publication of this article. This aspect is one of the main limitations of the study (as stated in the corresponding section of this research article).

### 2.5. Planning

In a similar way to how the NANDA taxonomy was used in the previous phase to name diagnoses, this part of the process involves using the NOC and NIC taxonomies to name goals and interventions, respectively [27]. This type of taxonomy is used to give a universal character to the care plan.

In the planning phase, there are four basic activities: determining priorities, setting expected outcomes (NOC) [28], choosing the interventions (IAS) [29] that need to be prescribed to achieve the objectives, and, finally, developing the care plan.

Once the care plan has been developed, it is expected that the proposed interventions will achieve the set objectives and thus solve the patient’s problems.

Within the interventions, in addition to nursing care, the patient could be treated by other health professionals to treat some alterations in a more specific way. For example, treating alterations in the locomotor system and their associated consequences via physiotherapy sessions and using psychological treatment strategies to treat alterations at a cognitive level would complement nursing interventions, offering the patient a specific and multidisciplinary treatment approach that would serve to improve their quality of life. All these theoretical aspects were applied to develop the care plan presented in this report. 

### 2.6. Evaluation

In this last phase of the Nursing Care Process, it is decided whether the goals have been achieved and whether the interventions are effective or whether any changes need to be made. In the evaluation, all steps of the care plan are reviewed to see if it has been carried out correctly. The outcomes of a care plan are the measurement instruments of the care plan. They can be positive (if the objective has been met), negative (if the objective has not been met), or unexpected (outcomes that appear unrelated to the interventions) [25].

### 2.7. Application of the Delphi Method

In the process of developing this standard and specific care plan for patients with Tay–Sachs disease, a report was presented as a starting point after a thorough analysis of the existing scientific literature. It addresses all aspects associated with this disease. A study of cases described from all over the world was carried out. In addition, in order to evaluate it, the care plan was subjected to the Delphi method.

The Delphi method is a structured communication technique used to gather opinions or judgments from a group of experts on a specific topic or issue. It consists of a series of rounds of questions and comments in which the experts provide their opinions or comments anonymously. The principal investigator compiles and summarises the responses and sends them back to the experts for further input. This process is repeated until a consensus is reached on the topic under study [30].

This process is often used in decision-making processes involving complex or uncertain issues, such as technology forecasting, risk assessments, or strategic planning. It is also used in fields such as the healthcare sector, where expert opinions are needed to make clinical decisions.

The advantages of the Delphi method include the possibility of gathering opinions from a geographically dispersed group of experts, the anonymity of the process, and the fact that it focuses on achieving consensus among the experts [31]. 

To validate our work, the Delphi method was applied, taking into account the following aspects:-Identification of the problem: First, the problem was defined, identifying the need to establish a specific and standard care plan for Tay–Sachs disease. It is important to note that no such care plans currently exist, and the repercussions that would derive from the establishment of a specific care plan would be positive for the scientific community in general.-Selection of the panel of experts: A group of experts on the topic in question is selected. These experts should have relevant knowledge and experience in the area and be able to provide informed answers. The selection of our group of experts was made through e-mail contact. Their profiles were carefully screened beforehand, taking into account their expertise in the field of rare diseases. The entry criteria for the group of experts requested was clinical, research, and teaching activity in the last five years in relation to the topic of rare diseases. Our final group of experts consisted of 9 healthcare professionals who are experts on the topic of rare diseases at a clinical, teaching, and/or research level and 1 person with expertise in the around-the-world movement of rare diseases. This person, who was an expert on patient associations, is a member of the Working Platform on Rare Diseases of the Province of Granada and was included in this process because of the perspectives she could offer due to her position.

Among the other nine professionals that were in our group of experts, five are nursing professionals in prestigious hospitals such as the University-Hospital Virgen de las Nieves and the University-Hospital of Jaén, one specialises in midwifery, one teaches and carries out research activity at the University of Alicante, one is a case manager at the Hospital San Cecilio, one is a podiatrist at the University of Malaga, one is a psychologist working in a research group on rare diseases, and one is a physiotherapist and an occupational therapist who conducts clinical and teaching activities at the Erasmus University Medical Center. The backgrounds of this group of experts were varied, and the group included both national and international members. All of the experts hold positions that are closely related to the world of rare diseases. As required by the method applied, the identity of the experts must remain anonymous.

-Creation of the care plan: The standard and specific care plan for patients diagnosed with Tay–Sachs disease was outlined by conducting an in-depth study of the scientific literature published on the subject and by thoroughly analysing the clinical cases described in terms of their clinical characteristics, phenotypes, and lifestyle. Once prepared, it was sent to the experts for them to study and provide comments. The care plan outline was accompanied by a series of clear and concise specific questions for the experts to answer. These questions addressed problems and doubts that the authors of the care plan encountered during the drafting process.-Sending of the care plan: the care plan was sent to the experts, who provided their answers anonymously.-Analysis of the responses: a detailed study of the responses was carried out to determine the degree of consensus among the experts. The analysis identifies areas of agreement and disagreement, as well as the reasons for discrepancies.-Resubmission of the care plan: a new care plan including amendments based on relevant input from the experts was drafted. Subsequently, issuing the care plan to the experts was again accompanied with asking them additional or revised questions based on the results of the previous analysis. This new report was sent to the experts for response.-Repetition of the process: The steps just mentioned are repeated until a satisfactory level of consensus is reached among the experts. Specifically, the process required three rounds to reach this consensus. These three rounds were carried out in response to each of the contributions offered by each expert. As mentioned above, the starting point is a basic report on which work begins. Questions aimed at improving the quality of the care plan were asked. Online meeting sessions were held in order to specify the details that needed to be discussed. It is important to keep in mind that the Delphi process is iterative, which means that it is repeated several times until an acceptable level of consensus is reached. In addition, it is critical that the results of the study be carefully interpreted and used appropriately.

The care plan presented in this article is the final one, the one that was developed after the three rounds of expert consultation.

The total evaluation process took place between February 2022 and March 2023. Communication with the group of experts was mostly hosted via email.

### 2.8. Ethical Aspects

For the development of this standard and specific care plan for patients diagnosed with paediatric Tay–Sachs disease, an in-depth study of the existing scientific literature was carried out, and the clinical cases described were analysed in more detail in terms of their clinical features, phenotypes, and lifestyle. The care plan was validated using the Delphi method.

Once the report has been published, it will be disseminated to hospital rare disease services and patient associations involved in Tay–Sachs disease.

At all times, the study was conducted in accordance with the guidelines of the Declaration of Helsinki, as amended by the 64th WMA General Assembly in Fortaleza, Brazil, in October 2013. The study was approved by the CEIM/CEI Province of Granada Ethics Committee with the following approval code: 02032021.

## 3. Results

### 3.1. Assessment

For Tay–Sachs disease, the most appropriate model is the Virginia Henderson [30] model, as it is a model based on the needs of a patient. The Virginia Henderson model emphasises basic human needs and is based on the idea that nursing should involve helping sick and healthy individuals to carry out activities that contribute to maintaining their health, recovering it in the case of health loss, or achieving a peaceful death. According to this model, the role of a nurse is to assist a sick or healthy individual in carrying out the activities they would carry out if they had the necessary strength, will, and knowledge. The author of this model describes the following concepts:

Person: A complex being (biological, psychological, and social) with 14 fundamental needs.

Health: The ability to function independently in relation to the 14 fundamental needs.

Environment: The set of external conditions that influence one’s health state and their development. These conditions can have positive or negative effects on the person. The nurse will strive to provide an environment favourable to health.

Care: The nurse supplements or assists the individual in performing the activities necessary to regain or maintain their state of health.

This theory identifies 14 basic and fundamental human needs that are shared among all human beings which may not be met due to illness or the fact that one has reached a certain stage of the life cycle; they may also be influenced by physical, psychological, or social factors.

For the assessment to be effective, the nursing staff must assess these 14 basic needs and the factors that may influence or modify them. To finalise the assessment, all data obtained should be organised according to the chosen model. Regarding Tay–Sachs disease, a nursing assessment for the disease is presented in Table 1.

### 3.2. Care Plan

The standard and specific nursing care plan for patients diagnosed with Tay–Sachs disease is presented in Table 2. 

Twenty-two standard and Tay–Sachs disease-specific problems were identified and are presented, including nursing problems, autonomy problems, and collaboration problems.

We also mentioned actual or potential problems in each section [27].

The identified problems that are the responsibility of nurses are referred to as nursing diagnoses [23].

Autonomy problems are those in which the patient’s independence is compromised [25]. They indicate a total or partial and/or temporary or permanent deficit in the patient’s physical or psychological ability to complete the relevant actions to meet their needs on their own. This type of problem will always be classified as real.

Collaborative problems are health problems in which the patient requires nurses to carry out the treatment and monitoring activities prescribed by another healthcare professional [23]. Such problems can be categorised as actual or potential.

## 4. Discussion

It is essential to understand the background in which the disease is developing in order to deepen the analysis in this study and come to the right conclusions.

In 1880, Warren Tay and Bernard Sachs [45] first described the most striking aspects of the disease. Visual degeneration and delayed mental development were the first two signs that led these two scientists to define the disease. They explained that this disease responded to a process of progressive neurodegeneration that occurred with epileptic seizures, blindness, paralysis, and death at an early age, with patients usually dying when they were no more than five years old. Later, in 1969, Shintaro Okada and Jhon O’Brien identified the enzyme responsible for the disease as Hexosaminidase-A [18]. At first, it was thought that the disease only affected Jewish communities and those at an early age, but as time went on, studies revealed that this was not the case.

The disease affected both people belonging to Jewish and non-Jewish communities, although the prevalence was always higher within Jewish communities [46]. On the other hand, cases of patients being affected by the disease outside of the childhood age range were described [15]. It is true that the disease is much more frequent and decisive in patients of that age, but this is not always the case [47].

Different approaches have been proposed to reduce the impact of this disease. On the one hand, there has been an increased focus on prevention and developing screening programmes to reduce the prevalence of this disease; on the other hand, there is an ongoing search for a definitive treatment option for patients with Tay–Sachs disease. Currently, there are no effective and curative treatments for this disease [8], despite the undeniable advances of gene therapy in this field [48,49].

Faced with this situation, the role of nurses takes on greater relevance as they are in charge of providing the necessary care to the patient, the family, and the environment and improving their well-being [23]. Nursing professionals require a series of specific indications to treat this type of disease, guaranteeing optimal quality of care.

It should be noted that this is a standard care plan, but before its implementation, it must be personalised and adapted according to the specific needs of each patient, modifying it, if necessary, after each assessment. This aspect was one of the first points of discussion raised by the group of experts, as the possibility of personalising and adapting the care plan according to the particularities of each case was presented as essential. The group experts stressed that the care plan reported in this study should never be viewed as an immovable dogma.

The nursing care plan presented here addresses the needs identified after the assessment, placing the patient at the centre of the therapeutic scheme and studying the interactions between the family and the environment with the appropriate perspective.

Due to the degree of dependency among Tay–Sachs patients, different diagnoses of autonomy can be observed; the need for help from a caregiver with some basic daily life tasks such as eating, dressing, or going to the bathroom should be evident [11].

Depending on this aspect, it would be advisable to assess caregiver fatigue, as it can be an important and influential factor in the development and evolution of the patient’s illness [17,50,51]. Considering the role of the caregiver and their relationship with the patient diagnosed with Tay–Sachs disease was a nuance mentioned by the group of experts since, in a situation of total or partial dependency, the state of the main caregiver can have a direct impact on the patient [52,53].

Impaired mobility entails various difficulties, increases the risk of falls, favours the development of pressure ulcers due to the couch–bed lifestyle, and has a negative impact on the development of associated musculature [13]. The podiatry and physiotherapy professionals within the group of experts placed special emphasis on these aspects and are responsible for proposing the objectives and interventions necessary to combat these problems [54].

Communication impairment and language delay affect the social interaction skills of this type of patient [55]. In addition, this factor has a direct impact on mental health, leading to the development of disorders such as anxiety and depression [46]. This was highlighted by the psychologist in the group of experts, who indicated that aspects related to the patient’s mental health should be included in this care plan.

The risk of infection, especially of respiratory origin, requires special attention in these patients. Airway patency and temperature control are of vital importance in the management of this disease [14].

The approach and treatment of this type of patient should be multidisciplinary and collaborative, taking into account all health professions that need to be involved.

Derived from this multidisciplinary character are collaborative problems in which nursing professionals need to cooperate with other health professions such as physiotherapy, psychology, pedagogy, and different medical specialties [25]. For this reason, we aimed to make the group of experts as diverse as possible, which ensured we gained perspectives from experts in multiple disciplines, taking into account the above-mentioned inclusion criteria and the specificity of the topic under discussion.

We must emphasise care relating to gait rehabilitation, maintenance of muscle tone, and respiratory physiotherapy as problems of collaboration with physiotherapists. The treatment of mental disorders and coping with their state of health and social skills can be complemented by collaboration with psychology professionals. Communication impairment and language delay are, of course, problems to be dealt with by collaborating with pedagogical professionals, and the diagnosis and management of different underlying diseases, as well as the evolutionary development of the disease, require collaboration with medical practitioners.

It should again be emphasised that this care plan is a standard care plan for this disease and that personalising and adapting the plan to the patient, their family, and their specific environment will be necessary before its implementation.

There were no major points of controversy among the experts who evaluated the report. The atmosphere throughout the process was one of debate and followed a philosophy of advancing and creating informed knowledge in order to develop the most comprehensive care plan possible. It is true that two aspects were discussed most insistently. On the one hand, the need to appoint a referral nurse for each patient was discussed. This would have been a nursing professional with the knowledge and leadership needed to apply the care plan, manage all interventions, and fulfil objectives while establishing communication with other healthcare professionals that make up the multidisciplinary team dealing with the particular case. On the other hand, it was emphasised that the care plan needed to be applied to a certain number of patients in order to finish testing it, but given the specific circumstances surrounding this type of disease, it was impossible to do both at once. It was determined that the best way to address this limitation was to make the care plan public, making it more visible by disseminating it in order to ensure that it reaches as many patients as possible. Once implemented, the plan should be monitored and possible improvements to the plan should be evaluated.

We encountered several limitations in carrying out this study, all of which were related to the specificity and scarcity of existing knowledge about this disease.

The formation of the group of experts was complicated since finding and bringing together people with knowledge of rare diseases in general and Tay–Sachs disease specifically was unlikely. The low prevalence of the disease and the geographic dispersion of diagnosed patients was also a factor that did not help in the formation of the expert group.

In addition, the paucity of existing knowledge published in the scientific literature on the syndrome also hindered the development of this work. Another limitation was encountered when trying to cover all the needs identified with the use of the diagnoses, outcomes, and interventions of the NANDA, NOC, NIC taxonomies; the addition of specific ones for patients diagnosed with rare diseases would be convenient. It is true that the proposed care plan has not yet been implemented, which has a negative impact on the study. Once published, widespread dissemination of the care plan is necessary to closely monitor the implementation of care and its impact on quality of life and to identify possible improvements.

This work has led to the discovery of new research directions that will be followed up on in the near future. One of these will be to evaluate the efficacy of the care plan after putting it into practice. Communication with the group of experts will be reestablished to disseminate, apply, and evaluate the results of implementation.

In addition, another line of research underlying this work is to study the psychoemotional aspects and the effects of this disease on the patients and relatives who suffer from it by means of a qualitative study.

## 5. Conclusions

The nursing approach to this disease currently lacks specific indications on care for the patient, family, and environment. The implementation of this care plan requires advancements in the knowledge of the disease in order to improve patients’ well-being.

The universalisation and standardisation of nursing care for this disease form a solid basis of knowledge for future, more in-depth lines of research on this subject.

## Figures and Tables

**Table 1 jpm-13-01222-t001:** Assessment according to the 14 basic needs of the Virginia Henderson model.

NEED	COMMENTS	SCALES
1. Breathing	Tay–Sachs patients have an abnormal breathing pattern due to the involvement of the muscles responsible for respiratory function. They have an excess of mucus that they are unable to expel on their own. Occasionally and depending on the degree of evolution of the disease, they need oxygen therapy. Most of the underlying infectious diseases are related to the respiratory system.	- Borg scale (Dyspnoea) [31]
2. Food/Hydration	Swallowing impairment in this type of patient is significant. Of particular note is the need for assistance with feeding and hydration.	- MUST [32] (Risk of malnutrition)
3. Elimination	Urinary and faecal elimination patterns are normal, although the use of absorbent pads is necessary due to reduced mobility. Assistance with respect to the use of absorbent pads (changing, cleaning, applying moisturising creams, monitoring for minor chafing or sores, etc.) is also necessary.	- Bristol scale [33] (Stool consistency) - Bonney test [34] (Urinary incontinence)
4. Mobility	Mobility is almost non-existent. Impairment of the locomotor system is one of the most evident symptoms of this disease. Hypotonia, ataxia, muscle weakness, spasticity, loss of balance, abnormal movements, and paralysis. Tay–Sachs patients should not be allowed to walk alone as there is a high risk of falls. The involvement of the locomotor system is one of the most disabling features of patients with this disease. In addition to the involvement of the locomotor system, there is the potential risk of blindness due to damage to the macula, which makes movement very difficult.	- Fall risk [35] (Risk of falling) - Tinetti [36] (Static and running equilibrium)
5. Rest/sleep	Rest and sleep pattern is not affected; it is usually the involvement of other needs that causes it to be altered. Breathing problems, problems related to elimination, or any symptoms of the disease can alter the patient’s rest pattern at any given moment.	- Oviedo [37] (Level of sleep satisfaction)
6. Dressing/undressing	Dressing, undressing, putting on shoes, and taking off shoes are parts of one’s daily routines that patients with this disease cannot perform on their own.	- Barthel [38] (Functional assessment) - Katz [39] (Autonomy activities of daily living) - Karnofski [40] (Quality of life)
7. Temperature	Temperature may be altered by infectious processes resulting from the disease.	-
8. Hygiene/Skin	Personal hygiene depends on the people around them. It is interesting to note the occurrence of pressure ulcers due to their lack of mobility.	- Norton [41] (Risk of pressure ulcers) - Braden [41] (Risk of pressure ulcers)
9. Security	These patients are unable to control their own medication. They experience total or partial impairments with respect to their cognitive functioning, sensory–perceptual skills, orientation, and attention. The epileptic seizures experienced by these patients, which can endanger the patient’s safety, need special attention and specific care in this situation.	- Pfeiffer [42] (Cognitive impairment)
10. Communication	The progressive deterioration of speech hinders communication. As cognitive functioning is impaired, language is also impaired. The general regression with respect to their social and mental skills prevents the patient’s level of communication from being normal. In addition, these people may suffer from deafness, which also impairs their ability to communicate.	-
11. Religion/Beliefs	Self-perception, health beliefs (i.e., knowledge of their disease(s), intervention(s), and admission), and the influence of life events on their personal state and religious values should be assessed. In patients with Tay–Sachs disease, assessing this factor is complicated as they are normally patients under five years of age, and at that age, it is very difficult to be aware of the disease. In order to assess this factor, it is advisable to consider their family and environment. On the other hand, the religious values of the patient are also studied for this factor, and the same problem is encountered. It should be noted that many of the patients suffering from this disease are of Jewish origin, so it is worth mentioning that religion is a very important pillar in their lives.	-
12. Work/Accomplish	The level of self-actualisation in Tay–Sachs patients is minimal as they are mostly dependent on the people around them.	- Duke-Unc [43] (Perceived social support) - Zarit [44] (Caregiver overload)
13. Recreational activities	Apathy is one of the symptoms of this disease, which influences the way they relate to their environment and spend their free time.	-
14. Learn	As described above, patients with Tay–Sachs disease have very advanced mental developmental impairments, which makes it impossible for them to learn properly and therefore affects their activities in daily life.	-

**Table 2 jpm-13-01222-t002:** Standard nursing care plan for Tay–Sachs patients.

DIAGNOSES NANDA	OBJECTIVES NOC	INTERVENTIONS NIC
(00074) Compromised family coping r/w *—The reference person is temporarily preoccupied and trying to manage his/her emotional conflicts and personal suffering and is therefore unable to perceive or act effectively with respect to the client’s needs. e/b *—The reference person displays behaviour that is disproportionate (over or under) to the patient’s capacities or need for autonomy.	(2600) Coping with family problems.	(7140) Family support. (7040) Support to the main carer. (5440) Increase support systems. (5270) Emotional support.
(00069) Ineffective coping r/w—Person’s vulnerability. e/b—Anxiety. Low self-esteem or chronic depression.	(2000) Quality of life.	(5820) Reduced anxiety. (5395) Improved self- confidence. (5880) Relaxation techniques.
(00108) Self-care deficits: bathing r/w—Musculoskeletal impairment e/b—Inability to wash body or body parts.	(0306) Self-care: instrumental activities of daily living (IADLs). (0301) Self-care: bathing. (0305) Self-care: hygiene.	(1801) Help with self-care: bathing/hygiene. (1805) Help with self-care: IADLs. (1804) Help with self-care: grooming.
(00110) Deficit of self-care: use of the toilet. r/w—Perceptual or cognitive disorder, muscular, musculoskeletal, etc. e/b—Inability to go to the toilet or complete potty training.	(0310) Self-care: toilet use.	(1801) Help with self-care: bathing/hygiene.
(00109) Self-care deficit: Dressing. r/w—Neuromuscular impairment. e/b—Impaired capacity to put on or take off necessary clothing.	(0302) Self-care: dressing.	(1802) Help with self-care: dressing/personal groooming.
(00102) Self-care deficit: Feeding. r/w— Neuromuscular impairment. e/b—Swallowing food.	(0303) Self-care: eating.	(1803) Help with self-care: feeding.
(00088) Impaired ambulation. r/w—Neuromuscular impairment. e/b—Impaired standing ability.	(0208) Mobility. (0202) Balance.	(0200) Promotion of exercise. (0221) Exercise therapy: ambulation. (0222) Exercise therapy: balance.
(00051) Impaired verbal communication r/w Central Nervous System Disorder. e/b Inability to speak	(0902) Communication. (0903) Communication: expressive. (0904) Communication: receptive.	(4976) Improving communication: speech deficit.
(00103) Impaired swallowing. r/w—Neuromuscular disorders, such as decreased gag reflex, decreased strength of muscles involved in mastication. e/b—Evident difficulty with swallowing.	(1010) Swallowing status. (1004) Nutritional status.	(1860) Swallowing therapy. (1803) Help with self-care: feeding.
(00046) Impairment of skin integrity. r/w—Physical immobility. e/b—Disruption of skin continuity.	(1913) Severity of physical injury. (1102) Wound healing: by first intention. (1103) Wound healing: by second intention.	(3584) Skin care: topical treatment. (3660) Wound care. (3520) Pressure ulcer care.
(00091) Impaired mobility in bed. r/w—Musculoskeletal impairment. e/b—Getting into bed from the supine and lying down position and vice versa.	(1909) Fall prevention behaviour.	(0840) Change of position. (0740) Care of the bedridden patient.
(00085) Impaired physical mobility. r/w—Musculoskeletal disorders. e/b—Musculoskeletal disorders. e/b—Musculoskeletal movement-induced tremors.	(0200) Ambulation.	(0200) Promotion of exercise.
(00131) Memory impairment. r/w—Neurological disorders. e/b—Inability to recall recent or past events.	(0908) Memory. (0901) Cognitive orientation.	(4760) Memory training. (4720) Cognitive stimulation.
(00052) Impaired social interaction. r/w—Limitations in social interaction. r/w—Limitations in social physical mobility and communication barriers. e/b—Observed inability to receive or convey a satisfactory sense of belonging.	(1610) Auditory compensation behaviour. (1611) Visual compensation behaviour. (1604) Participation in leisure activities. (0116) Participation in games. (1205) Self-esteem.	(4974) Improving communication: hearing impairment. (4976) Improving communication: speech deficit. (4978) Improving communication: visual impairment. (5100) Empowering socialisation.
(00061) Carer role fatigue. r/w—Increasing care or dependency needs. e/b—Concern about usual care.	(2508) Well-being of the main caregiver. (2603) Integrity of the family.	(7040) Support to the main carer. (5230) Increase coping.
(00155) Risk of falls. e/b—Impaired physical mobility and impaired mental status.	(1910) Safe home environment. (1908) Risk detection.	(6490) Prevention of falls. (8880) Environmental risk protection.
(00005) Risk of body temperature imbalance. e/b —Clinical processes affecting temperature regulation.	(1922) Risk control: hyperthermia. (1923) Risk control: hypothermia. (0800) Thermoregulation.	(3900) Temperature regulation.
(00004) Risk of infection. e/b—Chronic disease.	(1924) Risk control: infectious process. (1900) Vaccination behaviour. (1842) Knowledge: infection control.	(6550) Protection against infections.
(00032) Ineffective breathing patterns. r/w—Musculoskeletal impairment. e/b—Dyspnoea, shortness of breath, tachypnoea, cyanosis, nasal flaring, coughing, depth changes in breathing, etc.	(0403) Respiratory status: ventilation.	(3230) Respiratory physiotherapy. (3320) Oxygen therapy.
(00031) Ineffective airway clearance. r/w—Excessive mucus. e/b—Changes in respiratory frequency.	(1918) Prevention of aspiration.	(3160) Airway suction.
(00183) Willingness to improve comfort. e/b—Expresses desires to increase well-being and increase the feeling of satisfaction.	(3002) Patient/user satisfaction: communication. (3003) Patient/user satisfaction: continuity of care.	(5510) Health education. (4920) Active listening.
(00157) Willingness to improve communication. e/b—willingness to improve communication.	(1610) Auditory compensation behaviour. (1611) Visual compensation behaviour.	(4974) Improving communication: hearing impairment. (4976) Improving communication: speech deficit. (4978) Improving communication: deficit visual.

* Abbreviations are as follows: r/w—related with; e/b—expressed by.

## Data Availability

Not applicable.

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
