# Peer review of "Nursing Care Plan for Patients with Tay–Sachs—A Rare Paediatric Disease"

_jpm, 2023, doi:10.3390/jpm13081222_

Round 1

Reviewer 1 Report

Dear Authos,

This particular study is very interesting and the emerging results will improve of the current nursing treatment in mutlidisciplanery fields.

The manuscript is very well written, but it needs some small improvements, which I noted in the attachement file.

The introduction is very well written and contains everything the reader needs to understand the study.

Panel of experts: detail how the participants were selected and what their entry  criteria were. 

The discussion is well written but:

- the second paragraph (As described in previous sections,.....) is big enough and it is necessary to reduce it. It does not contribute anything important to the text and the study does not deal with religious believes.

-the fourth paragraph is duplicate 

 Τhe conclusions are well written.

Sincerely

Author Response

R1

Comments and Suggestions for Authors

Dear Authos,

This particular study is very interesting and the emerging results will improve of the current nursing treatment in mutlidisciplanery fields.

Thank you very much for your comments.

The manuscript is very well written, but it needs some small improvements, which I noted in the attachement file.

We will be pleased to attend to your requests.

The introduction is very well written and contains everything the reader needs to understand the study.

Thank you very much.

Panel of experts: detail how the participants were selected and what their entry  criteria were. 

This section is described in more detail.

The discussion is well written but:

- the second paragraph (As described in previous sections,.....) is big enough and it is necessary to reduce it. It does not contribute anything important to the text and the study does not deal with religious believes.

Right. We reduce the length.

-the fourth paragraph is duplicate 

Right. Duplicity is eliminated.

 Τhe conclusions are well written.

Thank you very much for your comments again.

Sincerely

Author Response

R2 Appraisal

 Nursing care plan for patients with Tay-Sachs and a rare paediatric disease.

 Summary: does not present results and conclusion. The methodology needs further clarification.

Thank you very much for your comments, all your requirements will be attended point by point.

Introduction:

  • Does not answer the object of It provides a detailed approach to this rare disease, but does not focus on the construction of the specific care plan for the person with this problem.
  • It states that "Nursing care is of vital importance for maintaining the quality of life and well-being of the patient, their family and the environment", but it does not talk about the nursing process, nor does it go into the need for a specific plan for these
  • Presents the central aim of this

A more connected objective is presented with the article offered and the need for the existence of this type of care plan is justified.

Method:

  • The method describes the nursing process. The nursing process is a problem-solving method used by nurses in their clinical practice. It is not a research method. Therefore, the authors should reformulate this entire chapter and present the method used, in line with the different international classifications. From reading the article, I understand that we are dealing with a consensus of experts, hence the use of the Delphi

More information is added on the design and methodology followed for this article.

  • The participants in the Delphi technique were The description of the participants states that "They are all closely related to the world of rare diseases". I think it should be defined what is meant by an expert. It is not enough to say that one is a rare disease carer. The criteria that determined their selection should be explicit. This is not evident in the study.

More information about the group of experts and their selection process.

  • In ethical terms, it complies with the guidelines for research

Thank you

  • The procedures have been described, but it is not sufficiently clear and objective about the aspects that were reformulated until We believe that there should be an organisation of questions that was guiding the answers of the experts/expertise.

The information required in the corresponding section has been expanded.

Results:

We agree with the presentation of the concepts introduced in this chapter. They are assumed as the metaparadigm of the philosophical perspective that was structuring the construction of this specific care plan for people with this rare disease.

Thank you very much for your comments

  • Table 1 is appropriate because it focuses on the 14 needs of Virginia Henderson's model and adjusts the assessment process to people with this rare

Thank you very much for your comments

  • The specific care plan is the culmination of this whole building process, involving all 10 I could not find evidence resulting from expert consensus in the field under study.

I understand that the arduous process of consensus building remains internally as field work for the authors' group. Normally the final report is presented although it is explained that it was evaluated using the Delphi method. In any case the evaluation process is available.

Discussion

It is from page 14, line 4 that the authors focus on the object of study. All of the above could be in the Introduction.

This aspect is corrected

The discussion needs more clarity and objectivity in relation to the problems identified.

The complete discussion is revised

It does not ensure an adequate interpretation/discussion, given the agreements and disagreements of the experts regarding the three aspects included in the care plan (Diagnoses, Objectives and interventions).

These requirements are answered in the discussion

References

  • There are 27 references prior to 2018 (more than five years old). There are 21 references from the last 5 And 5 of the references have no date.

The dates in the bibliography are revised.

  • It seems to us that this is insufficient for the quality of the journal in

Sorry for the discrepancy but working with rare diseases is quite complex. It cannot be treated as if they were other pathologies with more prevalence, really. Researching from a nursing point of view and on a specific rare disease is a real challenge. Finding 53 references on this subject, 21 of them from the last 5 years, believe me, is more than enough to support an article of these characteristics. I hope you will take this into account when making your next decision.

In view of the above, I consider that this article is not suitable for publication. I suggest its reformulation and new submission.

Thank you very much for your comments, I hope the changes made have improved the article

Reviewer 3 Report

The aim of this study is to present the nursing care plan for the maintenance of the quality of life and well-being of patients with Tay-Sachs, his family and environment, using Nursing Care Process 5 phases: Assessment, Diagnosis, Planning, Implementation and Evaluation.

Strengths:

-        Very few data in the literature concerning the topic

-        Using the Delphi method

-        Offering standardization of nursing care for this very rare pathology

Weakness:

-        Few elements describing the research methodology

-        lack of data on testing the proposed care plans

-     lack of data regarding the assessment of patients' quality of life (types of instruments used, etc.)

Please also give some details about the experts’ panel (eg. aria of expertise, etc.)  involved in the study and about the period of time and the place / conditions in which the research was carried out.

Author Response

R3

Comments and Suggestions for Authors

The aim of this study is to present the nursing care plan for the maintenance of the quality of life and well-being of patients with Tay-Sachs, his family and environment, using Nursing Care Process 5 phases: Assessment, Diagnosis, Planning, Implementation and Evaluation.

Strengths:

-        Very few data in the literature concerning the topic

-        Using the Delphi method

-        Offering standardization of nursing care for this very rare pathology

Thank you very much for your comments and for finding these aspects as strengths of the study. 

Weakness:

-        Few elements describing the research methodology

More information on the aspect you are requesting is included. Nursing methodology is actually applied throughout the care plan development process. The main source of information obtained was the analysis of scientific literature and the experience of the group of experts. This is now expressed in the report.

-        lack of data on testing the proposed care plans.

Right. This is one of the main limitations of the study and is expressed in the report. Once implemented, these issues will be followed up and addressed.

-     lack of data regarding the assessment of patients' quality of life (types of instruments used, etc.)

 Right. This is one of the main limitations of the study and is expressed in the report. Once implemented, these issues will be followed up and addressed.

Please also give some details about the experts’ panel (eg. aria of expertise, etc.)  involved in the study and about the period of time and the place / conditions in which the research was carried out.

The requested information is expanded in the corresponding section.

Round 2

Author Response

R2

Appraisal - 2nd version

Nursing care plan for patients with Tay-Sachs and a rare paediatric disease.

We agree with most of the changes made, particularly in relation to methodology and discussion.

Thank you again for your comments which are undoubtedly improving this report.

However, I must emphasise that:

In the method the nursing process is described. The nursing process is a problem- solving method used by nurses in their clinical It is not a research method.Therefore, the authors should reformulate this entire chapter and present the method used, in line with the different international classifications. From reading the article, I understand that we are dealing with a consensus of experts, hence the use of the Delphi technique. The care plan was constructed using a particular methodology. I suggest you consult the list of research methods, associated with levels of evidence (e.g. from the Joanna Briggs Institute).

Understood. The required information on the research method used in the methodology now appears.

  • I agree with what you have said about nursing methodology, but you should emphasise which method is

Right. The requested information now appears in the article.

  • I still believe that, given the quality of the journal, 50% of the references should be from the last 5 years and this condition is not

The bibliography has been updated and now more than 50% of the references are from the last 5 years.

  • The Abstract can still be improved according to these

The abstract is revised and updated

I hope I have answered all the requests and that finally the article can be published.

Thank you very much

Reviewer 3 Report

It will be interesting to continue your study by analyzing the results of implementation of this care plan, its impact on quality of life and to study possible improvements.

Author Response

R3

Comments and Suggestions for Authors

It will be interesting to continue your study by analyzing the results of implementation of this care plan, its impact on quality of life and to study possible improvements.

Thank you very much for your interest and support. Working in the world of rare diseases can sometimes be complicated. The email address of the corresponding author remains at your disposal should you be interested in following up on this line of research. Thank you again.